


# Assessment of building damages and adaptation options under extreme flood scenarios in Shanghai

Jiachang Tu[1,2], Jiahong Wen[1*], Liang Emlyn Yang[2*], Andrea Reimuth[2], Stephen S. Young[3], Min Zhang[1], Luyang Wang[1], Matthias Garschagen[2]

[1]School of Environmental and Geographical Science, Shanghai Normal University, 200234 Shanghai, China
[2]Department of Geography, Ludwig Maximilian University of Munich (LMU), 80333 Munich, Germany
[3]Geography and Sustainability Department, Salem State University, 01970 Salem, USA

*Correspondence to*: J. Wen (jhwen@shnu.edu.cn), L.E. Yang (emlyn.yang@lmu.de)

**Abstract.** Plenty of various measures have been taken to mitigate flood losses in Shanghai over thousands of years, including the construction of sea dikes and floodwalls. However, the combined effects of intensified rainstorms, sea-level rise, land subsidence, and rapid urbanization are exacerbating extreme flood risks and potential flood losses in the fast-developing coastal city. In light of these changes, this article presents an assessment of possible exposure and damage losses of buildings in Shanghai (including residential, commercial, workplace, and industrial buildings). Based on extreme flood scenarios caused by storm surges, precipitation, and fluvial floods, current flood-defence standards will soon be overtaken. Further analyses show that the inundation area could reach 9%, 16%, 24%, and 49% of Shanghai (excluding the area of islands) under the 1/200, 1/500, 1/1000, and 1/5000-year flooding scenarios, respectively. This study finds, in terms of the total building damage, the 1/5000-year flood scenario damage is more than ten times the 1/200-year flood scenario. Accordingly, the average annual loss (AAL) of residential, commercial, office, and industrial buildings are 13.9, 2.3, 5.3, and 3.9 million USD. Specifically, among the 15 (non-island) districts in Shanghai, Pudong has the highest exposure and AAL at all the four flood scenarios, while the inner city (including seven districts) is also subject to extreme AAL of up to 40% of its total building values. This study further addresses the possibilities of these extreme flood scenarios, and adaptation options such as: strategic urban planning, advanced building protections, and systematic flood management. Conclusions of the study provide information for scenario-based decision making and cost-benefit analysis for extreme flood risk management in Shanghai and is applicable to other similar coastal megacities.

## 1 Introduction

Coastal cities that have historically suffered from major flood impacts, such as: Dhaka, Guangzhou, Miami, Mumbai, New Orleans, and New York, are heavily affected by future rising flood risks to their population and assets (Woetzel et al., 2020a; Nguyen et al., 2021; Chan et al., 2021). By 2050, over 570 low-lying coastal cities and their 800 million inhabitants will face risks from the impacts of floods and rising sea levels, causing economic losses of up to 1 trillion USD (C40Cities, 2018). Frequent floods will also accelerate stresses of water supply (Yang et al., 2013), health care (Paterson et al., 2018), basin



management (Yang et al., 2018a), infrastructure maintenance (Yang, 2020; Bubeck et al., 2019), and ecosystem degradation (Tonkin et al., 2018).

Although many coastal cities are facing increasing flood risks (Liang et al., 2017), some will be more seriously affected than others due to local climate change effects, high exposure, or limited adaptation capacities. One of the hotspots, which is facing

high flood risks already, is Shanghai. Compared to other coastal cities, like Dhaka, Manila, and Rotterdam, Shanghai is the most vulnerable city to flooding (Balica et al., 2012). Coupled with factors of human (population exposure and property exposure) and environmental (low-lying land, land subsidence problems, the threat of sea-level rise, frequent typhoons and extreme precipitation) relationships, Shanghai is ranked as one of the top 20 cities to flooding in the world (Hallegatte et al., 2013; Wu et al., 2019). As an answer to this obstacle, Shanghai, the most developed financial centre in China, should

increasingly install flood protection, with a focus on hard measures.

Since the 1950s, hard measures, like seawalls (along the Yangtze River Estuary and Hangzhou Bay) and levees (along the Huangpu River), were constructed in Shanghai (Zhou et al., 2017). However, the seawalls and levees can be easily destroyed due to the lower structures and standards during the various stages of construction (Zhou et al., 2017). Only 23% of seawalls can withstand a 200-year flooding scenario; 58% of seawalls can withstand a 100-year flooding scenario while the rest of

seawalls withstand less than a 100-year flooding scenario (Wang et al., 2012). The present protection level of the levees along the Huangpu River for the lowest sections, according to the study, is around 1/50-year flooding scenario (Ke et al., 2018). Meanwhile, the historical crest heights in the Huangpu River reveals a significant growing trend from 1950 to 2000, with the existing levees failing several times and leading to extreme damage (Ke et al., 2018), posing a greater threat to Shanghai. Overall, to protect Shanghai from a future failure of floodwalls and levees, extreme flood scenarios, and their consequences

should be the taken as the first step of an integrative risk assessment.

Reviewing the current literature shows that various flood scenarios have been widely developed and validated for measuring flood risks in Shanghai. For example, according to the trend of relative sea level rise and the harmonic analysis of storm surges along the Shanghai coast, Yin et al. (2011) forecasts flood scenarios in 2030 and 2050. One two-dimensional MIKE 21 flow model has been applied in simulating future combined effects (sea level rise, land subsidence, and storm surges) of flood

scenarios in 2030, 2050, and 2100 (Wang et al., 2012). Fluvial floods from the Huangpu River were also simulated, considering land subsidence, sea level rise, and storm tide, in considering the return periods of 20, 50, 100, 200, 500, and 1000 years in Shanghai, respectively (Yin et al., 2013). Incorporated with three anthropogenic variables (land subsidence, urbanization, and flood defence), Yin et al. (2015) used a numerical 2D modelling approach for return periods of 10, 100, and 1000 years. In general, the flood scenarios produced in most existing studies tended to focus on the possible future flood scenario changes

rather than extreme events, e.g., the concern floods over a 1000-year return period.

The previous studies are dedicated to simulating floods or flooding in Shanghai instead of focusing on the overall risks. The consequences of flooding (i.e., construction damage), however, should be taken into consideration. Flood damage to buildings is noticeably a large part of the total flood losses in cities (Chmutina et al., 2014; Park and Won, 2019), in addition to the often devastating human costs (Woetzel et al., 2020a). In 1997, Typhoon Winnie caused large-scale floods in Shanghai and affected





more than 5000 households (Du et al., 2020). In 2008, floods damaged 160 streets and 13,000 residential buildings in Shanghai. Therefore, local communities and governmental departments are increasingly calling for holistic analyses of possible building damage under extreme flood scenarios in order to accurately understand and assess potential flood impacts (Kelman and Spence, 2004). Accurate loss data play an integral role in assessing the damages of buildings. But obtaining accurate data is a challenge shared in many areas (Middelmann-Fernandes, 2010), especially in assessing the damage of buildings. To estimate

building stocks and the values at risk under a 1/1000-year extreme flood scenario, Wu et al. (2019) integrate census-level building floor-area data and geo-coded building asset value data. Shan et al. (2019) further assessed the flood losses of residential buildings and household properties in Shanghai based on a stage-damage function, building footprint, and housing prices. Mostly, deeper investigations into the uncertainties (e.g., asset values, damage rate, and flood process) with reliance on flood risks and damages of buildings (e.g., residential, commercial, office, and industrial) are also urgently needed in order to

better support decision making which enhances the overall flood resiliency of the city.

To address these questions, we adopted the very extreme flood scenarios with return periods of 200, 500, 1000, and 5000 years. The four extreme flood scenarios are assumed as integrative effects of multiple flood-triggering factors, such as typhoon-induced storm surge, precipitation, fluvial flood in combination with a high astronomical tide, to reflect low probability-high impact flood situations in Shanghai. The objective of this paper is to assess flood losses of residential, commercial, office, and

industrial buildings under extreme flood scenarios in Shanghai. To achieve the objective, we modeled building exposures at four extreme flood scenarios with return periods of 1/200, 1/500, 1/1000, and 1/5000, respectively. Combining the exposure maps with the stage-damage functions, the study evaluated and identified the spatial distribution of losses for the specific types of buildings in Shanghai. Section 2 of the paper introduces the data and details of the methods, and section 3 presents the data analysis and major results of the study. Section 4 discusses future flood scenarios and proper adaptation strategies for building

a flood-resilient Shanghai. Final conclusions are described in section 5.

## 2 Data and Methods

### 2.1 Study Area

Shanghai is the biggest coastal city in China in terms of population (24.3 million in 2019) and is the major trading and financial hub of China. The city has an area of 6340.5 km$^2$ that lies in the Yangtze River Delta along the northern edge of Hangzhou

Bay (Figure 1). Shanghai is prone to flooding because of its flat low-lying terrain, as well as its location on the path of frequent typhoons from the northwest Pacific (Balica et al., 2012). Moreover, the city experienced an average land subsidence of 1.97 meter from 1921 to 2007 and the trend is continuing (Gong and Yang, 2008) additionally driving flood risks (Quan, 2014). In addition, the total building area in Shanghai reached 1368.8 km$^2$ (SMBS, 2018) which includes residential buildings (686.5 km$^2$), commercial buildings (81.6 km$^2$), workplace buildings (90 km$^2$), industrial buildings (283 km$^2$), and others. The total

construction industry value is 159.8 billion USD (SMBS, 2018).





Figure 1. Satellite image-based map of the study area (Shanghai's 15 districts, excluding the city's islands) (The satellite image background is from Esri).

## 2.2 Data

The study involves the data of four extreme flood scenarios, building maps, land use and land cover data, and the construction costs of different buildings in Shanghai. In this study, we adopted the very extreme flood scenarios with return periods of 200, 500, 1000, and 5000 years. Based on the extreme water levels for different return periods, the hydrodynamic modelling is composed of atmospheric models (Fujita typhoon model), ocean models (TOMAWAC, TELEMAC), and coastal models (MIKE 1D/2D), developed to simulate four extreme flood scenarios, respectively combined with a fluvial flood during

Typhoon Winnie in 1997 (Wang et al., 2019). Typhoon Winnie brought the highest recorder water level with 5.72 meters since 1900, which caused the collapse of 148 meters of floodwalls and overflowed 57 km of floodwalls and 69 km of sea dikes. The rainfall and river discharge data based on Typhoon Winnie in 1997 are superimposed on coastal flood simulations. These four flood scenarios are raster data with a 60-meter spatial resolution.





The data for Shanghai's buildings were acquired from Baidu Map (Baidu Maps, 2017) using a python-based web crawler, and
then processed with ArcGIS software. Baidu Maps provide various map services, such as satellite images, street maps, and
route planners in China. The shapefile data for Shanghai's buildings include the information of building type, building ground-
based area, height, and the number of floors for each building. These data are used in combination with land use data to further
cluster the buildings into four different types including: residential, commercial, workplace, and industrial.

The land use data of 2013 from the Shanghai Planning and Land Bureau was depicted hierarchically into 3 sectors, 15
subcategories, and 73 subclasses of land use types using ArcGIS 10.6.1. The reclassification of the land use was conducted
according to the National Standard of China "GB/T 2010-2017" (CSP, 2017) and covers residential area, commercial area,
workplace area, and industrial area. Specially, the land use type of residential areas covers apartment, mixed apartment,
housing in the rural, and empty housing. Commercial areas are the lands that are used for commercial operations. Workplace
areas include land for medical care and health, charity, education, culture, government, research, market, and insurance.
Industrial factories are classified into industrial areas.

The cost data of building construction used in this study are derived from the 2019 annual report provided by the consulting
company Arcadis in Shanghai. The cost per square meter is based on Construction Floor Areas (CFA), which measures to the
outside face of the external walls. The data set depicts four different building types (domestic, office/commercial, hotels,
industrial, and other), which are further divided into 19 subsectors (Arcadis, 2020). Construction costs of various buildings are
averaged in order to get the mean value for each building type (table 1).

Table 1. Common construction costs of various buildings in Shanghai.

| Building Type | | Construction Cost (USD/m$^2$ CFA) | Average Construction Cost (USD/m$^2$) |
|---|---|---|---|
| Residential | Apartments, high rise, average standard | 668 - 740 | 873 |
| | Apartments, high rise, high end | 1554 - 1697 | |
| | Terraced houses, average standard | 446 - 477 | |
| | Detached houses, high end | 666 - 740 | |
| Commercial | Retail malls, high end | 1228 - 1585 | 1157 |
| Office | Medium/high rise offices, average standard | 868 -1156 | 1406 |
| | High rise offices, prestige quality | 1158-1445 | |
| Industrial | Industrial units, shell only (Conventional single story framed units) | 432 -540 | 486 |

The relationship between flood inundation depth and flood loss of a building or other property is depicted by a stage-damage
function (Garrote et al., 2016; Mcgrath et al., 2019). Based on actual building damage data from past flood hazard events and
previous empirical stage-damage functions in Shanghai (Yu et al., 2012; Wang, 2001), Ke (2014) developed updated stage-
damage functions to specific buildings in Shanghai (Figure 2). These stage-damage functions represent the generalized loss of
one type of buildings with similar properties, which are adopted in the present study.
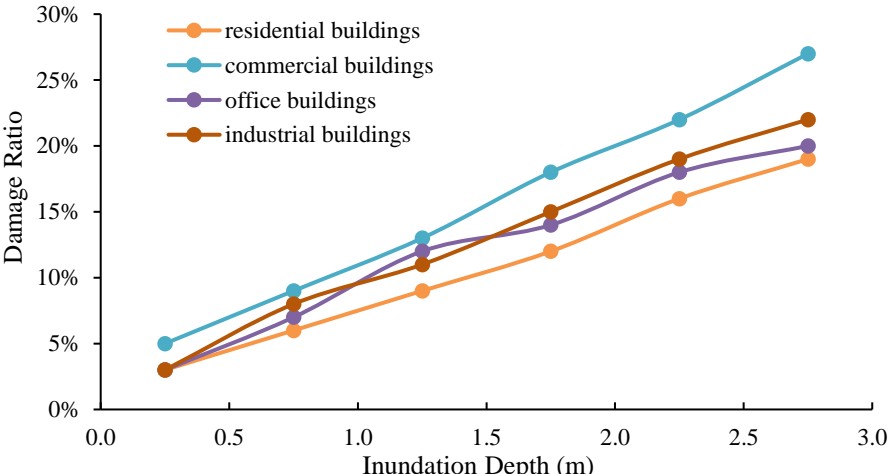

Figure 2. Stage-damage functions for buildings in Shanghai (Adopted from (Ke, 2014)).

## 2.3 Methods

In our study, the flood damages in Shanghai are estimated within three different steps: first, we calculated the asset values for each building based on the surface area of the building and the average construction cost. Then, the exposed building areas are determined by overlaying the distribution of buildings with the inundation maps for the four extreme flood scenarios. The damage values of a building could be estimated based on the exposed building area and the stage-damage functions. Finally, the overall flood risk for buildings in Shanghai, described as the estimated average annual loss (AAL), is calculated using a polynomial regression analysis. More details of the assessment are introduced in the following sections 2.3.1 to 2.3.4.

### 2.3.1 Asset value analysis of buildings

Different methods exist in the evaluation of building values. According to the Chinese National Standard (GB/T 50291-2015) and previous studies, there are four main approaches to evaluate the asset value of buildings: method of the sales comparison, method of the income capitalization, method of the construction cost, and method of the hypothetical development (CSP, 2015). The application of different methods depends on the specific study aims and the availability of building-specific data. Method of sale comparison is often used when the evaluation of a similar type of building is available. Method of income capitalization is suitable for buildings that yield profits like rents. If a building is newly constructed or (to be) reconstructed after damage, method of construction cost would be suitable for the evaluation. Method of hypothetical development is applicable to the evaluation of real estate with investment development or redevelopment potentials. Since the present study aims to assess flood damages on buildings, the construction cost method is used with consideration of the building surface area. Then, the asset value of one building can be approximated by the following function:

$$W_n = S_n \times P_n \tag{1}$$





Where $W_n$ (USD) is the asset value of buildings for building type $n$, $S$ is the surface area of building $n$, $P_n$ is the construction

cost (USD/m$^2$) for the specific type of building $n$.

### 2.3.2 Evaluation of building damages in flood

The water depth of the flooding scenario determines the exposed area of the buildings. If the building is flooded at depth of

more than 3 meters, we assume that the exposed area covers two floors instead of one. The potential building damages are

determined by the stage-damage functions. The stage-damage function is used to evaluate the damage values of residential,

commercial, workplace, and industrial buildings with the probability of 1/200, 1/500, 1/1000, and 1/5000-years extreme flood

scenarios, respectively. The specific stage-damage functions are derived from existing studies on the relationship of various

building loss rates with water-level depth in Shanghai (Penning-Rowsell et al., 2013; Ke, 2014). The damage values of one

building can be expressed by the following function:

$$f(x) = E_n \times P_n \times T_n \tag{2}$$

Where $f(x)$ represents building damage for building type $n$, $E_n$ represents the exposed area of buildings for building

type $n$, $P_n$ is the construction cost (USD/m$^2$) for the specific type of building $n$, $T_n$ represents the damage proportion from

stage-damage function for building $n$ under different water-level depths.

### 2.3.3 Integrative building damages

When expressing a city-scale flood damage for different flood scenarios, we use the already well established economic AAL

(Hallegatte et al., 2013). The AAL is the sum of the probabilities of the floods for each return period, while considering the

approximate areas under the associated risk curve (Ward et al., 2011). In the present study, we considered only the damage

value of buildings as a major part of the AAL. Particularly, the AAL represents the integrative building damage for all types

of buildings in all the considered flood scenarios in Shanghai. Hence, we get the AAL values for different exceedance

probabilities (extreme flood scenarios) as the sum of:

$$\text{AAL} = \sum_{x=rp_{min}}^{rp_{max}-1} \big(f(x) + f(x+1)\big)/2 \times \Delta x \tag{3}$$

Where n is the building type, $x$ is the return period of the flood scenario, $f(x)$ is the damages value of one building.

### 2.3.4 Spatial pattern identification

In this study, the AAL of all sub-districts and their neighbours were compared with the AAL by Getis-Ord Gi* in ArcMap

10.6. The results contain a significant range of high values (hot spots) and low values (cold spots). Hot spots mean the AAL

of a sub-district has a high value and can be surrounded by other sub-districts with high values as well. Cold spots indicate an

opposite situation.





# 3 Assessment of building damages in extreme floods

## 3.1 Mapping the flood scenarios

This section presents the comparison of the spatial distribution of inundation areas under the four extreme flood scenarios
(Figure 3). The inundation areas increase generally, but non-linearly, along with increasing return periods. As shown in figure
3, the inundation areas mainly concentrate on regions around the Wusongkou (the mouth of Huangpu River), the Suzhou creek
mouth (the central of Shanghai), the Songjiang-Qingpu low-lying area, and the north bank of Hangzhou Bay. The inundation
areas expand to inland areas from the coastal region and the rivers as the flood scenario becomes more extreme. For the 1/200-
year flood scenario, 9% (488 km$^2$) of Shanghai is flooded, mainly along the north bank of Hangzhou Bay. The flooded area of
Shanghai increased to 16% (868 km$^2$) and 24% (1302 km$^2$) in the 1/500-year and 1/1000-year flood scenarios, respectively.
The flooded area extended significantly to 49% (2659 km$^2$) of Shanghai in the 5000-year flood scenario. More specifically in
this exceptionally extreme scenario, 32% of the city would be flooded with a water depth of 0-0.5m, 11% with a 0.5-1.0m
depth of water, and 5% with a water depth of more than 1m (Figure 3).

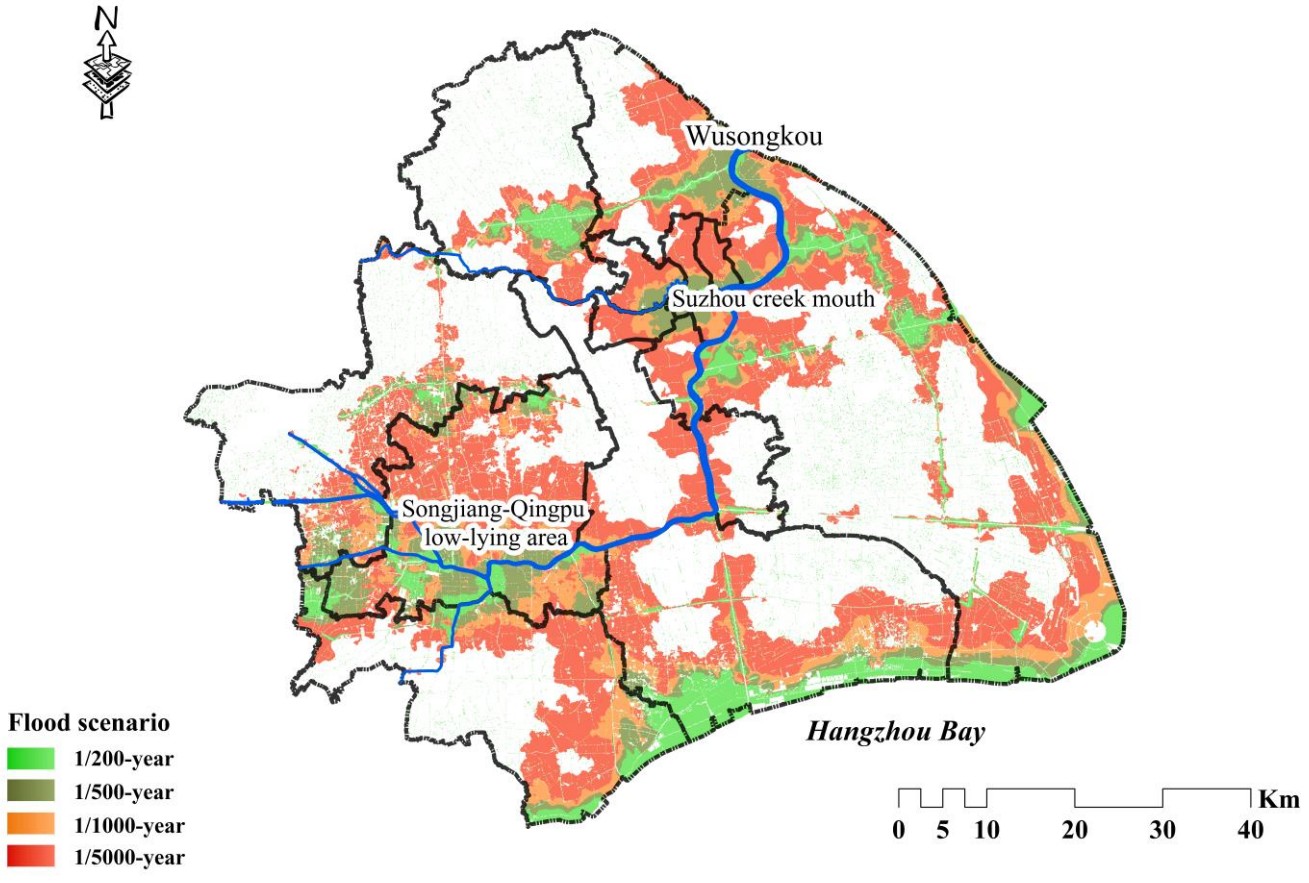

Figure 3. Inundation areas for the four flood return periods in Shanghai (excluding the area of islands).

## 3.2 Estimating the building assets

This study identified the building floor area (BFA) of each type of building in 2017, which amounts to 816 km$^2$, 52 km$^2$, 152 km$^2$, and 300 km$^2$ respectively for the residential, commercial, workplace, and industrial buildings. Accordingly, the building asset values are 2494, 212, 747, and 510 billion USD considering the Average Construction Cost (see Table 1) of each building

types in 2019. This is a conservative estimation as the actual present value of the buildings are certainly higher due to new developments in recent years. The relative numbers show that residential buildings have the highest asset value, followed by office, industrial, and commercial building.

Figure 4 shows that all types of buildings have higher values within the outer ring of Shanghai. This is consistent with the fact that the central downtown area is densely developed with the accumulation of various buildings. While comparing the building

asset values in the 15 districts, the Pudong district has the highest asset value and largest building surface area (Table 2) as it is the largest district in Shanghai.

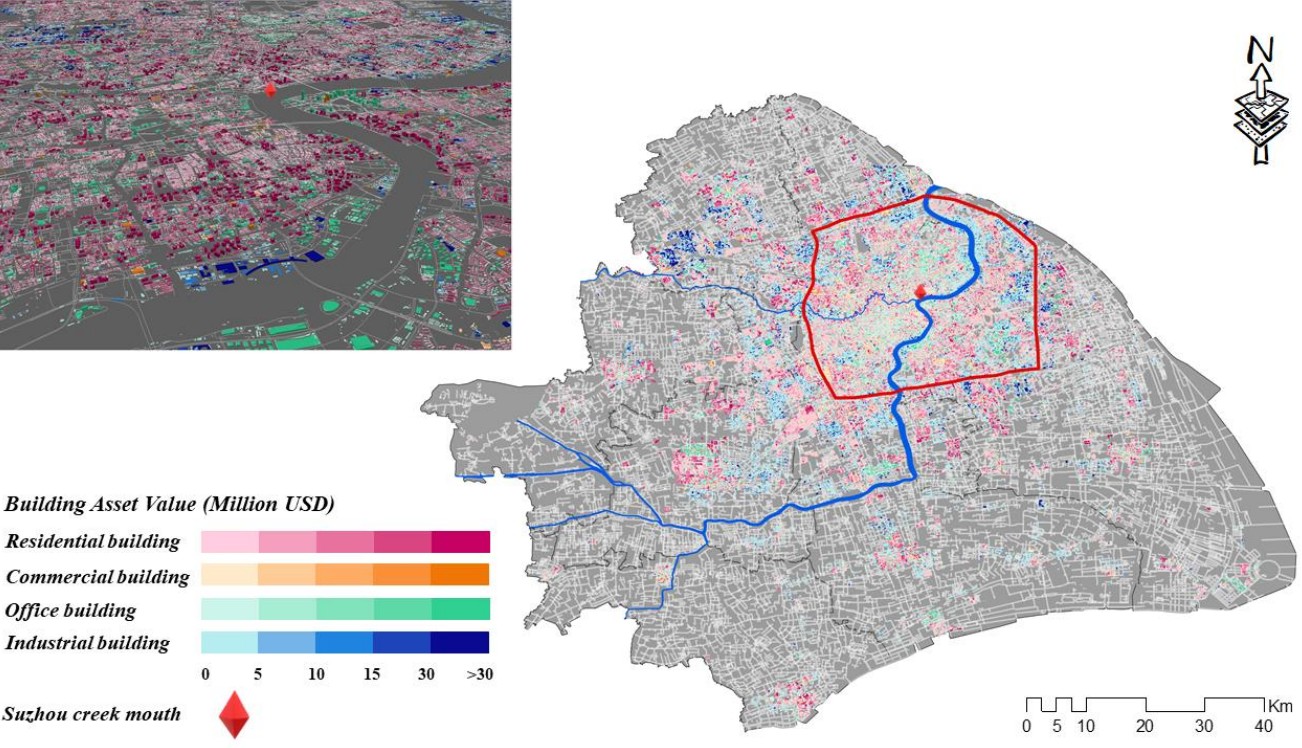

Figure 4. The distribution of building asset values in different types of buildings in Shanghai.

Table 2. The building asset values and floor areas in the 15 districts of Shanghai.

| District | Residential | | Commercial | | Office | | Industrial | |
|---|---|---|---|---|---|---|---|---|
| | Asset Value (Billion USD) | Surface Area (km$^2$) | Asset Value (Billion USD) | Surface Area (km$^2$) | Asset Value (Billion USD) | Surface Area (km$^2$) | Asset Value (Billion USD) | Surface Area (km$^2$) |
| Jiading | 236 | 271 | 25 | 22 | 54 | 39 | 89 | 183 |
| Fenxian | 64 | 73 | 9 | 8 | 17 | 12 | 8 | 16 |



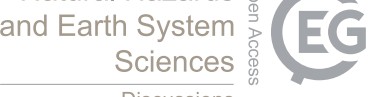

| Baoshan | 260 | 298 | 22 | 19 | 48 | 34 | 64 | 131 |
|---|---|---|---|---|---|---|---|---|
| Xuhui | 117 | 133 | 10 | 9 | 58 | 41 | 16 | 33 |
| Putuo | 133 | 152 | 13 | 11 | 28 | 20 | 18 | 37 |
| Yangpu | 90 | 104 | 6 | 5 | 58 | 41 | 22 | 45 |
| Songjiang | 217 | 249 | 12 | 10 | 40 | 29 | 27 | 55 |
| Pudong | 616 | 705 | 51 | 44 | 200 | 142 | 164 | 336 |
| Hongkou | 72 | 82 | 3 | 3 | 22 | 16 | 4 | 7 |
| Jingshan | 42 | 48 | 6 | 5 | 8 | 6 | 2 | 4 |
| Changning | 59 | 68 | 1 | 1 | 25 | 18 | 4 | 9 |
| Minhang | 303 | 347 | 35 | 30 | 69 | 49 | 58 | 119 |
| Qinpu | 128 | 146 | 6 | 5 | 59 | 42 | 15 | 31 |
| Jingan | 96 | 110 | 7 | 6 | 32 | 23 | 18 | 37 |
| Huangpu | 61 | 70 | 5 | 4 | 28 | 20 | 3 | 5 |
| Total | 2494 | 2857 | 212 | 183 | 747 | 531 | 510 | 1050 |

## 3.3 Exposed building values

The quantitative assessment and mapping generated the spatial extent of exposed buildings and the exposed building values under the four extreme flood scenarios in Shanghai (Figure 5). The assessment shows that the total exposed building values reach 39, 107, 166, and 386 billion USD under a 1/200, 1/500, 1/1000, and 1/5000-year flood scenarios, respectively. Exposed buildings in the 1/200 scenario are located mainly along the coast and rivers, with some exposed buildings in dispersed low-lying places. With an increase of extreme flooding, the exposed areas rise rapidly and become contiguous. Under the most extreme scenario of a 1/5000-year return period, the region around the Suzhou Creek Mouth in the inner city is remarkably exposed in deep water with the highest exposed values. Two contiguous inundation areas, central Shanghai and the Songjiang-Qingpu low-lying area in west Shanghai, are identified as the most seriously exposed regions in the most extreme flood scenario (Figure 5d).

Unsurprisingly, as the residential buildings have the highest asset value, their exposed values are also the highest in the four types of buildings (Appendices Table A1). Further analysis of the exposed values in different districts indicates that Pudong has the highest exposed value for all four flood scenarios (Table 3). The exposed ratio of each district in different flood scenarios is as follows: in the 1/200-year flood scenario, Fengxian has the highest percentage of exposed building values; under the 1/500-year and 1/1000-year flood scenarios, Huangpu has the highest percentage of exposed building values. Under the 1/5000 flood scenario, Hongkou has the highest percentage (Appendices Figure A1).

Water depth is a determinate factor of exposed building values. Though hardly visible in Figure 5, 94% of the exposed buildings are exposed to water levels of 0-0.5m in the 1/200-year flood scenario (Figure 5a). The account declines to 82%, 83% and 67% under the 1/500, 1/1000, and 1/5000-year flood scenarios, due to the increasing proportions of exposures in deeper water. In the case of the most extreme 1/5000-year flood scenario, 24% of exposed buildings are in water of 0.5-1m, and 8% of the exposed buildings are flooded in depth of 1-1.5m (Appendices Figure A2).

Natural Hazards
and Earth System
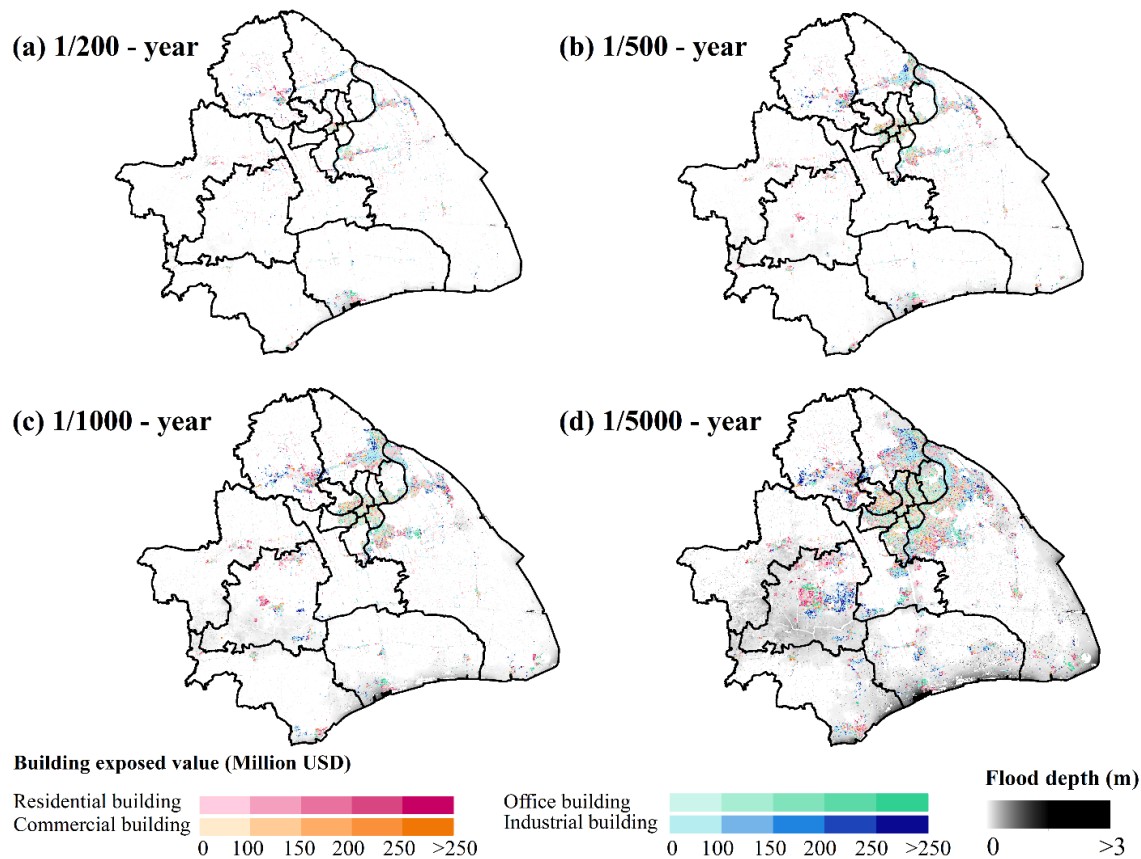

Figure 5. The distribution of exposed asset values for residential, commercial, office, and industrial buildings under four extreme flood scenarios in Shanghai.

Table 3. Exposed building values of the 15 districts in Shanghai under the four flood scenarios (Unit: Billion USD).

| Districts | Flood scenarios (Return Periods) | | | |
| --- | --- | --- | --- | --- |
| | 1/5000 | 1/1000 | 1/500 | 1/200 |
| Jiading (JD) | 23.27 | 11.48 | 9.00 | 5.96 |
| Fengxian (FX) | 9.69 | 4.03 | 2.93 | 2.69 |
| Baoshan (BS) | 43.52 | 25.87 | 17.39 | 2.61 |
| Xuhui (XH) | 23.43 | 7.19 | 4.79 | 0.70 |
| Putuo (PT) | 23.22 | 11.32 | 6.76 | 0.27 |
| Yangpu (YP) | 24.75 | 8.51 | 3.89 | 0.36 |
| Songjiang (SJ) | 35.67 | 9.54 | 3.48 | 0.96 |
| Pudong (PD) | 111.16 | 46.79 | 31.95 | 17.41 |
| Hongkou (HK) | 15.53 | 5.04 | 3.02 | 0.05 |
| Jinshan (JS) | 8.19 | 3.69 | 2.29 | 1.20 |
| Changning (CN) | 13.66 | 8.28 | 4.97 | 0.12 |
| Minhang (MH) | 15.56 | 0.93 | 0.55 | 0.62 |





| | | | | |
|---|---|---|---|---|
| Qingpu (QP) | 5.24 | 2.62 | 1.93 | 1.06 |
| Jingan (JA) | 19.41 | 9.47 | 8.13 | 3.02 |
| Huangpu (HP) | 13.69 | 11.56 | 5.73 | 1.96 |
| Total | 385.98 | 166.31 | 106.80 | 38.99 |

## 3.4 Damages of buildings in floods

The quantitative assessment provides maps of flood damages to different buildings under the four extreme scenarios (Figure 6). The total building damages in the 1/5000-year flood scenario is 18 billion USD, which is more than 10 times the 1/200-year flood scenario (1.39 billion USD). Again, residential buildings are the most damaged in all four scenarios with a damage value of up to 9.6 billion USD in the 1/5000-year scenario. Damages of industrial buildings, office buildings and commercial buildings would reach 1.6, 4.0, and 3.0 billion USD respectively under the 1/5000-year flood scenarios.

The damage analysis of different districts shows that Pudong has the highest overall damage in all scenarios (Table 4). However, the rankings of the proportion of damages are different. For the 1/200-year scenario, Fengxian has the highest proportion of the asset damages. Jingan has the highest damage proportion in the 1/500-year scenario. While for the 1/1000 and 1/5000-year scenarios, Huangpu is ranking as the most damaged district in terms of the damage proportion (Appendices Figure A3).

In terms of damage due to water depths in different flood scenarios, water depths of 0-0.5m causes 83 percent of building damage in the 1/200-year flood scenario. Under 1/500, 1/1000, and 1/5000-year flood scenarios, the proportion are 67%, 68%, and 44 %, respectively, with deeper water depths causing more damage. For instance, 35% of buildings are damaged in water depths of 0.5-1m under the 1/5000-year flood scenario, and 17% of the damages occur in water depths 1-1.5m (Appendices Figure A4).



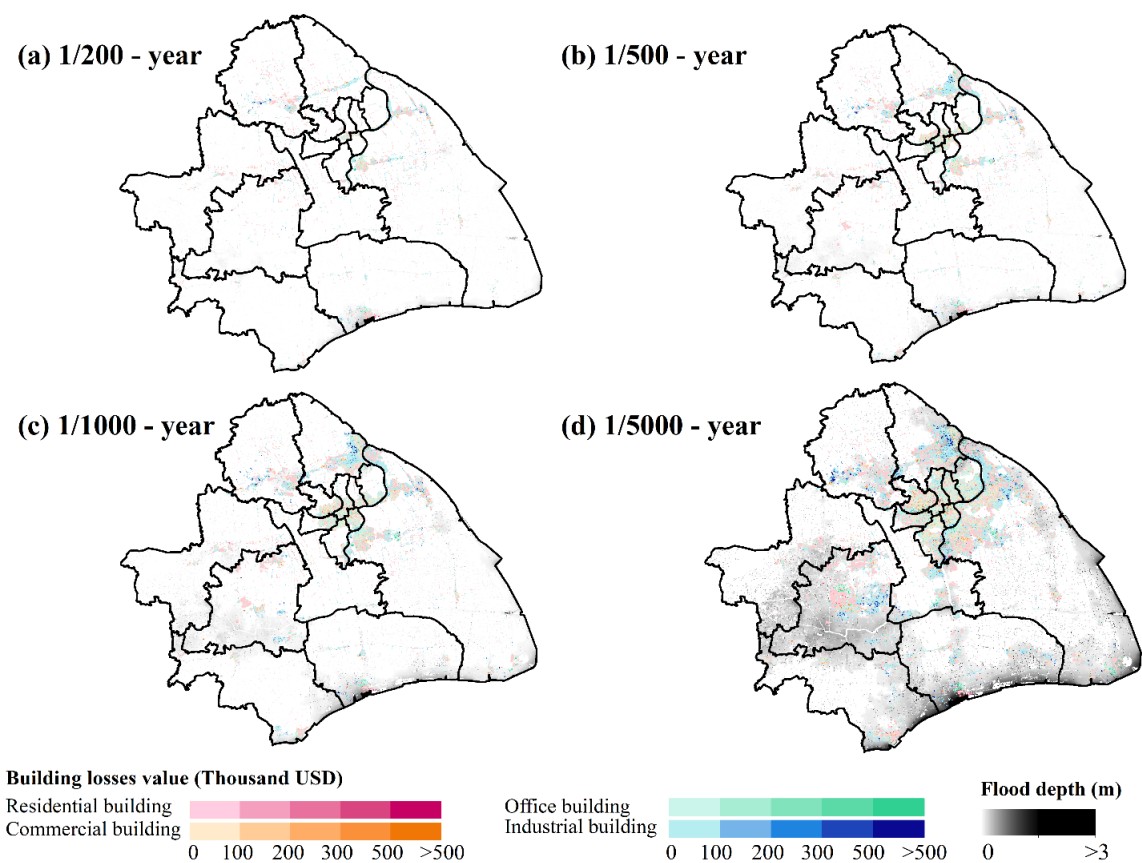

Figure 6. The distribution of damaged asset values for residential, commercial, office, and industrial buildings under the four extreme flood scenarios in Shanghai.

Table 4. Statistics of damage to residential, commercial, office and industrial buildings to extreme flooding under four return
period scenarios in districts (Unit: Billion USD).

| Asset Damages District | Return Periods | | | |
|---|---|---|---|---|
| | 1/5000 | 1/1000 | 1/500 | 1/200 |
| Jiading (JD) | 0.81 | 0.42 | 0.30 | 0.20 |
| Fengxian (FX) | 0.55 | 0.28 | 0.20 | 0.17 |
| Baoshan (BS) | 2.42 | 1.04 | 0.58 | 0.09 |
| Xuhui (XH) | 1.07 | 0.31 | 0.17 | 0.02 |
| PuTuo (PT) | 1.24 | 0.53 | 0.26 | 0.01 |
| Yangpu (YP) | 1.15 | 0.34 | 0.14 | 0.01 |
| Songjiang (SJ) | 1.31 | 0.33 | 0.11 | 0.03 |
| Pudong (PD) | 4.74 | 1.71 | 1.10 | 0.59 |
| Hongkou (HK) | 0.78 | 0.24 | 0.12 | 0.00 |
| Jinshan (JS) | 0.37 | 0.16 | 0.09 | 0.05 |
| Changning (CN) | 0.82 | 0.42 | 0.22 | 0.01 |





| | | | | |
|---|---|---|---|---|
| Minhang (MH) | 0.54 | 0.05 | 0.02 | 0.02 |
| Qingpu (QP) | 0.20 | 0.10 | 0.06 | 0.03 |
| Jinan (JA) | 1.23 | 0.62 | 0.46 | 0.09 |
| Huangpu (HP) | 0.95 | 0.53 | 0.26 | 0.06 |
| Total | 18.18 | 7.08 | 4.10 | 1.39 |

### 3.5 Integrative evaluation of flood damages in Shanghai

By integrating the extreme flood scenarios and associated building damages for the four types of buildings, we plotted the four
average annual probability-damage curves (Fig. 7). The AALs of residential, commercial, workplace, and industrial buildings
are 18.3, 3.6, 7.8, and 5.8 million dollars, respectively. It is clear that residential buildings would suffer the highest damage
value among the four types of buildings.

By using the Getis-Ord Gi* statistic tool (Hot Spot Analysis) in ArcMap 10.6, the results reveal the distuibution of high and
low building damages for different types of buildings in the community level in Shanghai (Fig. 8). Apparently, the distribution
of hot spots and cold spots for different types of buildings are quite different. For residential buildings, there are six hot spot
areas and five cold spot areas. Most of the hot spot areas concentrate in the ciy center except for one along the north coast of
Hangzhou Bay. Further analysis of the commercial buildings indicates a significant hot spot south of the Suzhou Creek Mouth.
Figure 7c shows five hot spot areas and two cold spot areas for office buildings with three hot spots located in coastal areas.
Four hot spots of industrial buildings concentrate mainly in the north, while the city center is the main cold spot area because
few industrial buildings are located there.

Overall, the city center is the hot spot area of flood damages for the residential, commercial and office buildings (Figure 8a,
8b and 8c). But in contrast, the city center is the cold spot area for the industrial buildings (Figure 8d). Wusongkou is a hotspot
for four different types of structures. Wusongkou floods in all four flood scenarios, and the inundation area expands as the
retrun-period expands. Another reason is that the density of building in Wusongkou is higher than in other areas.

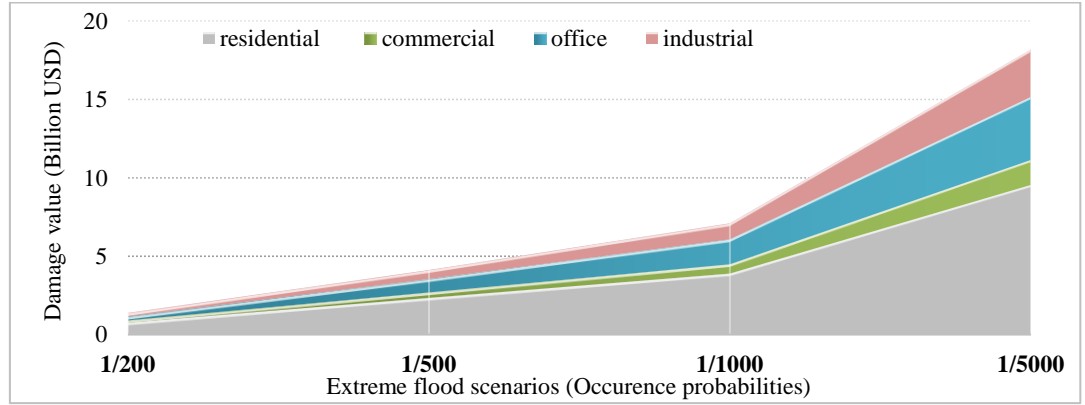

Figure 7. Stage-damage curve of extreme flooding for residential building, commercial building, office building, industrial
building and the sum of residential, commercial, office and industrial buildings in Shanghai.





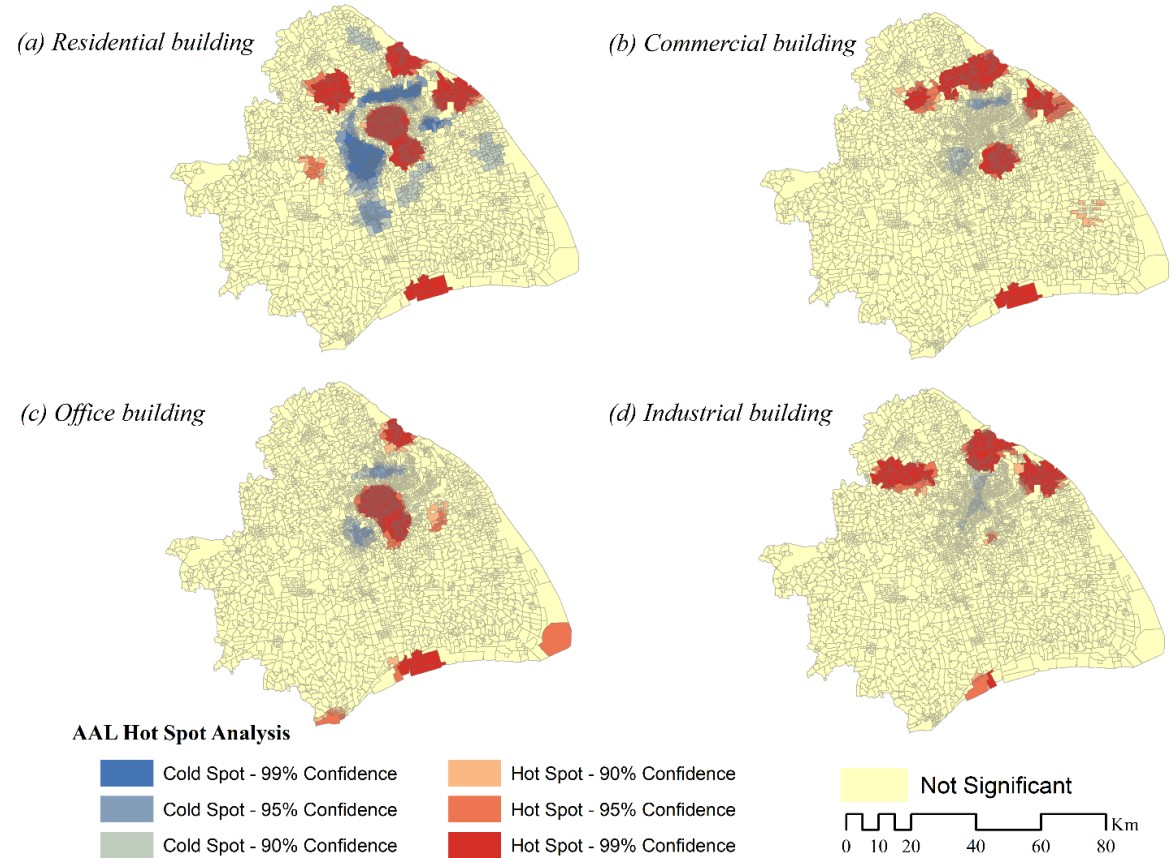

Figure 8. The hot spot analysis of the AAL for buildings at sub-district level in Shanghai.

## 4 Discussion

### 4.1 Evaluation of the flood risk in Shanghai and implications for the future

Our study shows that the damage to buildings in Shanghai grows exponentially with the decreasing likelihood of extreme flood scenarios. For instance, the resulting flood damages to residential, commercial, workplace, and industrial buildings under the 1/5000-year flooding scenario is more than ten times the resulting damages from a 1/200-year flooding scenario. As shown in section 3.1, the area along the Yangtze River Estuary, Hangzhou Bay and Huangpu River are broadly flooded under the 1/200, 1/500, 1/1000, and 1/5000-year flooding scenarios. The results of the study show the importance of assessing the risk to extreme events on regional scale at a high spatial resolution considering the differences in the exposed assets. The hot-spot clusters are distributed over the whole study area and vary from building type to building type. In some areas, the damage is driven mainly by high inundation depths (e.g., the hotspot in the south), whereas other areas face a high risk due to the high





vulnerability of the asset values. This shows the importance of assessing the different drivers of risk on local scale for the
selection and dimensioning of adequate protection measures against extreme events.

Concerning increases in climate change, the frequency and/or severity of acute climate hazards and the intensification of chronic hazards will increase the flood risks in Shanghai in the future (Woetzel et al., 2020b). The Sixth Intergovernmental Panel on Climate Change (IPCC) found that global precipitation will intensify and become more frequent in most regions with additional global warming (IPCC, 2021). Extreme precipitation events increased dramatically by 10% to 20% every 10 years
during the 1951 - 2001 period in the Yangtze River basin, China (Wang and Zhou, 2005). Concerning Shanghai, after analysing Shanghai's hourly precipitation records (1916 - 2014), Liang and Ding (2017) found a rate increase of 1.5 and 1.8 for heavy precipitation events. Precipitation events now increase the possibility of seawall and levee failures in Shanghai. One 1/1000-year return period flood occurred in Shanghai in 2013, breaking the highest crest record at Wusongkou Datum and causing levees to breakdown (Ke et al., 2018). As a result of climate change, extreme flooding events will become more common in
Shanghai.

Human activities can also increase the likelihood of flood risk in Shanghai. For instance, changes in land subsidence relative to the sea level rise could increase the flood risk to Shanghai. Due to the extraction of ground water, construction of high-rise buildings and underground projects (Gong et al., 2008), the average annual rate of land subsidence was 7 mm between 2007 and 2010 (Wu et al., 2012), and then the rate decreased to 5 mm/year after 2010 (Yin, 2011). However, the maximum annual
subsidence rate in Shanghai could still have a chance to reach 24.12 mm/year (Wang et al., 2012). On the other hand, sea levels will rise a maximum of 86.6 mm, 143mm, 185.6 mm, and 433.1 mm by 2030, 2040, 2050, and 2100, respectively in Shanghai (Wang et al., 2012; Wu et al., 2012). Future flood damage in Shanghai will be exacerbated by increased precipitation, land subsidence and sea level rise, which further shows the need to adapt to the (currently) low probability-high impact events.

## 4.2 Adaptation strategies to extreme floods in Shanghai

Effective adaptation to increasing flood risks requires an integrated climate response strategy, which shall include a broad scope of intervention measures such as urban planning, structural flood management measures, early warning systems, nature-based solutions, flood awareness and risk financing instruments (Yang et al., 2015; Jongman, 2018). Urbanization as a confirmed trend in the fast-developing coastal city may increase asset exposures to floods, but can also offer opportunities for improving flood risk management (Garschagen and Romero-Lankao, 2015). A top-down urban master plan, including land
use planning, control of runoff, access to data and information, etc. should be updated, by the Shanghai Municipal Government, to involve advanced risk management measures (Zhou et al., 2017). For instance, in its Master Plan 2017 - 2035, Shanghai is going to further develop its five new district centers at Jiading, Songjiang, Qinpu, Fengxian and Nanhui. These five district centers are planned to be nodal areas in Shanghai and provide more public services for the growing population. However, based on the flood scenario maps, the Songjiang-Qingpu low-lying area is a hot spot of flood damages. Therefore, future flood
protections in these locations, particularly the drainage system and the building structures, must be designed to a higher standard.





The hard, soft and hybrid measures must be considered in implementing the planned urbanization process (Table 5). It has been widely proven that the hard strategies can effectively reduce the flood hazard probability. For instance, the direct economic flood loss significantly decreased after a series of integrated flood management followed a mega-flood across central and south China in 1998 (Bryan et al., 2018). The protection level of existing seawalls and levees along the Changjiang Estuary, Hangzhou Bay and Huangpu River do not provide adequate protection to meet Shanghai's current flood defense standards. These structures are not sufficient to protect the increasing urban assets considering the combined impacts of climate change, land subsidence and typhoon events.

Table 5. Comparison of flood adaptation measures in Shanghai

| Categories of measures | Specific measures | Effects | Suitability |
|---|---|---|---|
| Hard measures | Seawall and levees | Protects areas from being flooded or eroded by extreme storms, floods, astronomical tides, and sea level rise, particularly in low lying areas. | Raise levees and construct a flood barrier in Wusongkou which could lower the flood pressure from Huangpu River (Wang et al., 2011). |
| | Drainage system | Rapid rainfall discharge to improve transport and safeguard property. | The drainage system should protect Shanghai under 50-year flood scenario. The capacity should be enhanced to the probability period of 100 years in the vulnerability (UPLR, 2018). |
| | Reservoir | Save part of the precipitation and reduce flood pressure in downstream or lowland areas. | Adjust stock by season or weather forecasting, relieve the pressures of the city flood management during floods. |
| | Channel | Relieve the flood pressure and speed up drainage within the city. | Ensure that the dam functions could be running on the Huangpu River and Suzhou River. |
| Soft measures | Warning system | Enable stakeholders or households to prepare for the extreme climate and react to mitigate it. | Could be used in Shanghai, especially preventing people from putting their lives in extreme flood event. |
| | Dry proofing | Being watertight with all elements substantially impermeable to the entrance of flood and with structural components having the capacity to resist flood loads (FEMA, 2013). | Help the household, especially for household that have experienced regular flooding, to stop the floodwater from their entrance door. |
| | Wet proofing | Allow water to enter the building but minimizing damage. | Minimizing the damage of property in the building in the flood-prone area. |
| | Detention and retention areas | Alleviate flood peak by artificially made storage areas (Glavan et al., 2020) | Based on Sponge City project, to capture, purify and store more water (Griffiths et al., 2020). |
| | Emergency relief | Personnel evacuation and transfer of property from short-term extreme precipitation. | Adapted in Shanghai, especially residents who lived in inundation areas in the extreme flood scenario. |
| | Insurance | Increase financial resilience to floods (Surminski and Oramas-Dorta, 2014). | Meet the needs of vulnerable individuals, households and micro, small, and medium-sized enterprises (MSMEs) in Shanghai (Hess and Fischle, 2019). |
| | Wetlands | Provide valuable flood storage, buffer storm surge, and assist in erosion control (EPA, 2021). Absorb and slow down the floodwater from storm surge. | The wetland on the north of Hangzhou Bay, the mouth of Yangtze River should be protected to slow down the erosion from the storm surge and sea level rise. |






Particularly for the hot spot areas where huge damages are expected, extra coping strategies must be taken seriously into consideration. For instance, underground water storage and pumping facilities are necessary for the city center, and these shall be systematically planned together with detention and retention regions on the ground. A feasible and practical way is to make use of the old air defense facilities and some underground parking lots (Chen et al., 2018). Researchers have mentioned, and

we also recommend, to enhance the bank and build a water wall in the estuary area of the Huangpu River (Chen et al., 2018), because of the large population and assets which deserves a high level of protection. Developing such a flood barrier and upgrading the Huangpu-River floodwalls to a protection standard of a 1/1000-year flood event in Shanghai could significantly reduce the expected annual flood damages to 0.07 - 0.5 billion/year in 2100 for the RCP (representative concentration pathway) 4.5 scenario (Du et al., 2020).

Potential flood risks coming from extremely low-probability storm surges can be further reduced by combining the soft strategy. For instance, dry proofing and wet proofing, as well as the coastal wetland strategies can be combined to form a hybrid strategy in specific regions. Although soft measures on their own cannot maintain the future flood risk at a low level, they can play a critical role in reducing potential damages. Additionally, the soft measures such as maintaining coastal wetlands can enhance social welfare by providing multiple ecosystem services. Challenges may rise in that homeowners are on average

not inclined to install soft strategies due to the high costs for individual households, even if they live in flood plains (De Ruig et al., 2020). On the other hand, extreme flood scenarios often cause more serious flood damage to households because they occur very soon and leave less time for responding (Yang et al., 2018b). Thus, an efficient forecasting and early warning system is needed which could help individuals, businesses, communities and government leave dangerous zones, transfer house property and improve preparedness with sufficient lead time (UN, 2020). Insurance also plays a very important role in

the support of various people and groups to recover from flood hazards, especially for the high-risk regions.

### 4.3 Uncertainties and limitations of the assessment

The integrative analysis of geospatial building asset maps, flood scenarios and the stage-damage functions in the study makes it possible to assess the flood damage of buildings in the mega city Shanghai with a high spatial resolution. However, the accuracy of building asset values could still be improved. First, the adopted building data of location, footprint area, height and floors didn't consider the construction materials used and years built. Also, the data from 2013 is not very new, considering

the fast development of Shanghai. Second, the classification of different types of buildings is quite straightforward based on the land use/land cover data. However, many buildings with multiple functions (e.g., shopping mall and offices) were identified to a single building type, which causes uncertainties of the building value. Third, existing studies' stage-damage functions for specific types of buildings are used to create the asset building loss map for the flood risk assessment. The functions must be

updated and tailored to more current and specific building conditions, particularly when estimating flood damages in the future (Ke, 2014).

Our assessment of the building damages is comparatively less than those in similar studies of Shanghai. The major reason is that we adopted the construction cost as the values of different buildings, while many other studies calculated the market value





of buildings and the associated properties. For instance, Wu et al. (2019) estimated the exposed building asset value in Shanghai
to be roughly 304.1 billion USD with a total damage of approximately 32.2 billion USD for a 1/1000-year flood event. Shan
et al. (2019) estimated the loss of the residential buildings to be 27.1 billion USD using house price data under a 1/1000-year
flood event in Shanghai.

The four extreme flood scenarios in Shanghai were taken from published models in Shanghai that are induced by atmospheric,
hydrology, and coastal models (Wang et al., 2019). However, the climatic impact-drives (CIDs) such as the frequency of sea
level rise, heavy precipitation, and pluvial floods can be altered as a result of various greenhouse gas (GHG) emissions. For
example, every 0.5°C rise in global warming increases the severity and frequency of heat extremes, such as heatwaves and
heavy rains, as well as agricultural and ecological droughts in some regions (IPCC 2021). Therefore, the CIDs (e.g., sea level
rise, heavy precipitation) should be considered in future flood simulation models and studies.

## 5 Conclusion

This study presents an integrated impact model of flood damage to buildings based on extreme flooding scenarios in Shanghai.
The results show that the inundation area is significantly larger in the low probability of extreme flood scenarios. In all the
four considered scenarios, the areas near the Huangpu River and along the shore are always the affected with building damages.
The central downtown areas of Shanghai have a high risk of being exposed to and affected by extreme floods, partly because
of its high building density. Besides that, the Songjiang-Qingpu low-lying area in the west of Shanghai has been recognized
as a noticeable area to be flooded under a 1/5000-year flood scenario. This calls for special concerns in the near future because
the Songjiang-Qingpu area is planned to become an important sub-center node.

Residential buildings account for the most damage of the four types of buildings, accounting for 47 percent of the total,
followed by industrial, workplace, and commercial buildings, respectively. The total asset value for the four building types is
3963 billion USD while the total AAL is 22.2 million dollars. It is also noticeable that the total damage for the four types of
385    buildings is 18 billion USD in 1/5000-year scenario, residential buildings are significantly vulnerable to extreme flooding in
contrast to the three other types of buildings.

The presented method offers the possibility to estimate the damage values for residential, commercial, workplace, and
industrial buildings in Shanghai under extreme flooding. It increases the accuracy and details of flood damage estimates for
different types of buildings by considerging the direct damage of buildings. The dynamic linkage between the extreme flooding
390    scenarios and the distribution of asset values of the four different types of buildings allows the evaluation of the spatial
distribution of flood damages, which would be valuable for individual real eastate managers and also for the city government.




**Appendices**

**Table A1.** Exposed values of different types of buildings under the four flood scenarios in Shanghai (Unit: Billion USD).

| Building types | Flood scenarios (Return Periods) | | | |
| --- | --- | --- | --- | --- |
| | 1/200 | 1/500 | 1/1000 | 1/5000 |
| Residential building | 22 | 63 | 97 | 217 |
| Commercial building | 3 | 6 | 10 | 25 |
| Office building | 7 | 22 | 35 | 78 |
| Industrial building | 7 | 16 | 25 | 66 |
| Total | 39 | 107 | 166 | 386 |

395

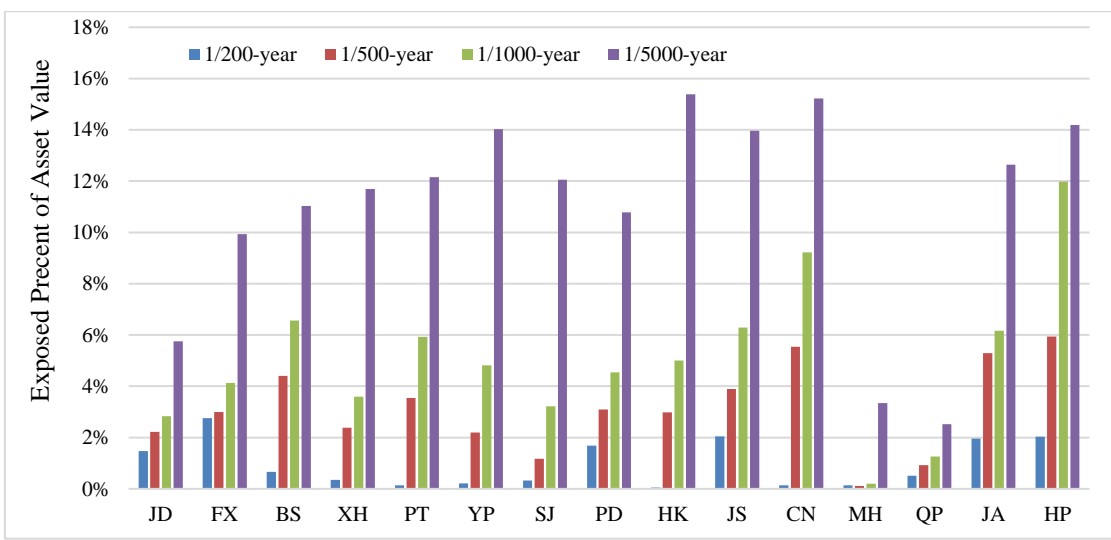

**Figure A1.** Exposed percentage of residential, commercial, office and industrial buildings to extreme flooding under four return period scenarios in each district.





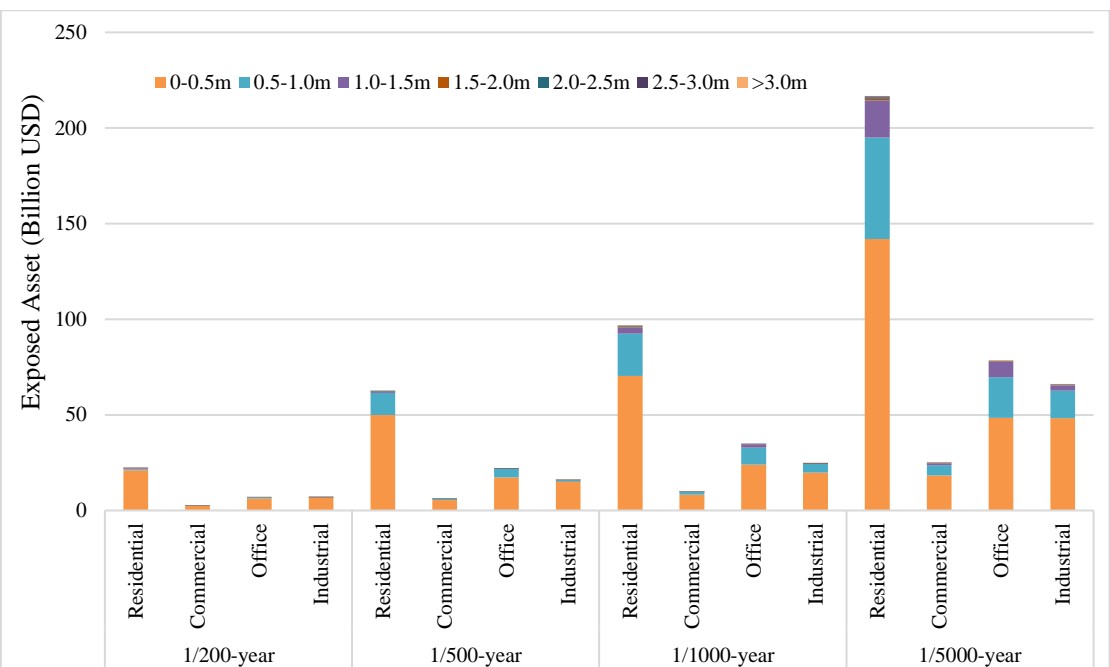

**Figure A2.** The exposed asset categorized by different inundation depths.

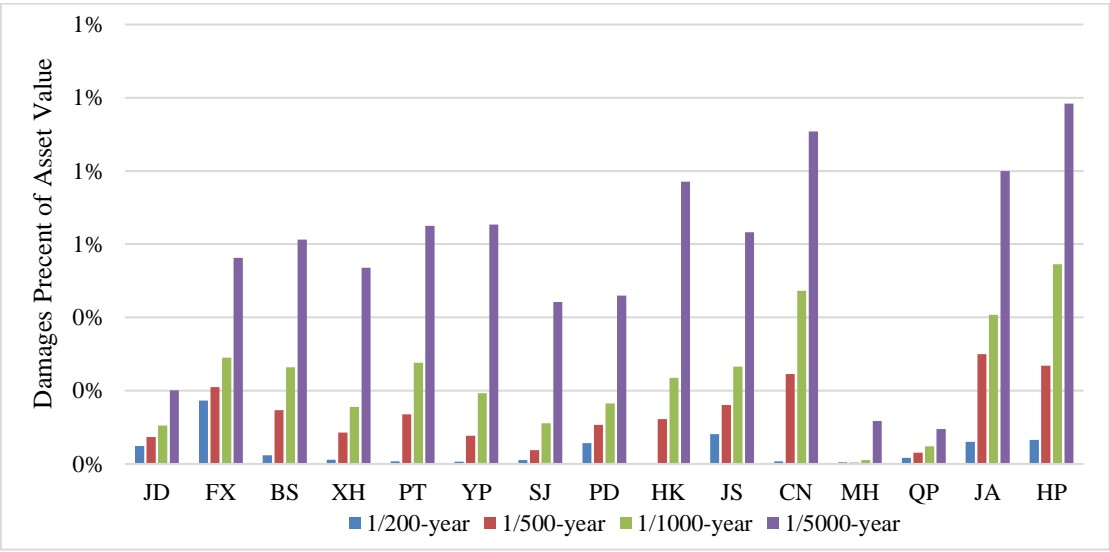

**Figure A3.** Damage percentages of residential, commercial, office and industrial buildings to extreme flooding under four return period scenarios in each district.

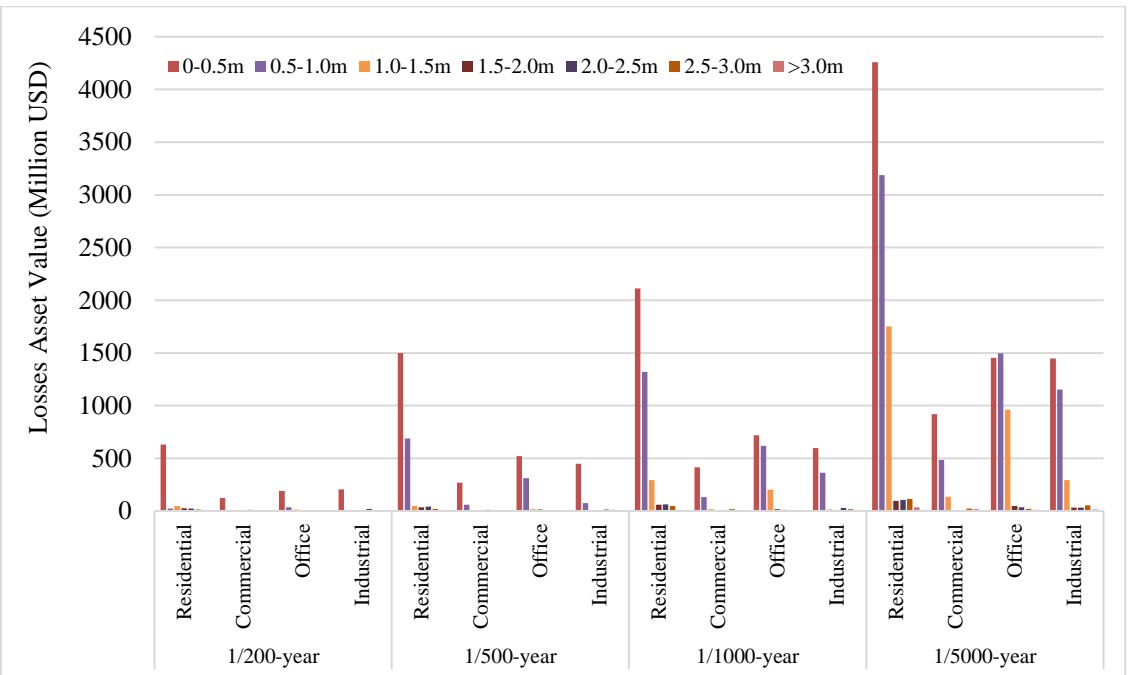

**Figure A4.** The losses asset in different water depths.

## Data availability

Data used in this study are available from the first author upon request.

## Author Contributions

J.T. analyzed the data, conceived and wrote the paper; J.W. and L.Y. conceived and co-wrote the paper; A.R., M.G. and S.Y. reviewed and improved the analysis and manuscript; M.Z. and L.W. provided the simulation results of the extreme flood scenarios.

## Funding

This research was funded by the National Natural Science Foundation of China (Grant No. 42171080, 41771540, 41871200) and the National Key Research and Development Program of China (Grant No. 2017YFC1503001).



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
