# Peer review of "Assessment of building damages and risk under extreme flood scenarios in Shanghai"

_Natural Hazards and Earth System Sciences, 2021_

## Author Response (AR1)

| | Comments | Submitted Answer |
|---|---|---|
| | | Review 1 |
| General comments | Thank you for inviting me to review the manuscript entitled 'Assessment of building damages and adaptation options under extreme flood scenarios in Shanghai'. This manuscript assesses possible exposure and damage losses of buildings in Shanghai and provides a detailed description of the technical methods and results using the case study. It is well written, and the results are clearly presented. However, my primary concern is about its theoretical or methodological contributions to the field of flood risk assessment, which are not sufficiently articulated or developed. Assessing very extreme flood scenarios (e.g., return period = 5,000 years) is not innovative enough by itself. | Thank you so much for pointing out that the theoretical or methodological aspects of the flood risk assessment are not sufficient. To introduce the methodology more clearly, we re-constructed and rewrote the paper. We are deleting the section "data and methods" and adding the new sections "Study area" (p.7. Line 87-95) and "Materials and methods" (p.8. Line. 98-194). In the new section "Materials and methods", we included one graphic at the beginning of the section to clarify our methodological procedure (p.8. Line 105). In order for readers to grasp our equation, we provide two tables as samples to present the calculation process: 1) 'building asset value' (p.13. Line 160); 2) 'the damage values' (p.15. Line 177).

To answer the reviewer's concern that the return period of 5,000-year flood scenario is not the innovation. The flood scenario map itself is not an innovation, and we explain the innovation of this study as follows: (1) The integrated flood model that simulated extreme compound flood scenarios was first produced in Shanghai. We developed an integrated numerical simulation system, coupling the atmosphere (Fujita typhoon model), oceanic/coastal environment (TELEMAC model), and river discharge (MIKE model) to simulate the compound extreme flood events. The simulation system was employed to provide four inundation scenarios of the inundation water levels as a function of 1/200, 1/500, 1/1000, and 1/5000-year, respectively in Shanghai for the first time. (2) The building flood risk in this study is a clear enough future for Shanghai. Although this study doesn't integrate future scenarios (e.g., representative concentration pathway, shared socioeconomic pathways, etc.), data from 2013 (building shapefile data, land use data) and 2019 (construction cost data) generate results that reflect the flood risk challenge under the current physical and socio-economic situation in Shanghai. The result links directly to disaster risk management, imply the extent of flood risk in building types, districts, and communities to the Shanghai Master plan, references result to future climate change scenario framework, information for scenario-based decision making, and cost-benefit analysis for extreme flood risk management in Shanghai. (3) The building flood damages, risk, and risk's spatial patterns were for the first time evaluated in the whole city (except the islands) on fine scale (community and district level). |
| Other general comments | Why is it needed to assess extreme flood scenarios with return periods of 5,000 years? | Thank you for this point. It is necessary to have flood risk assessment on such a low probability-high impact scenario that is increasingly possible to happen due to three reasons. According to the IPCC report, extreme seal level rise events are projected from once per century to once per year (IPCC, 2019) which could increase the frequency of extreme flood events for coastal cities, such as Shanghai. Shanghai may suffer from extreme compound flood threats in the next few decades considering risks from typhoons, sea level rise, heavy precipitation, and riverine flows due to its physical environment and location. Second, Shanghai currently relies extensively on hard measures of flood protection. But the seawalls and levees can be destroyed because of the multiunit |

| | | constructions and standards used during the long construction process. The seawalls and levees can't protect Shanghai from extreme flood events especially considering the fast population growth and social economic development that aggregate flood risk. We have enhanced the description of this point in "Section 1 Introduction" (p.3. and p.4. Line 36-61). |
|---|---|---|
| | Please can you provide more information about what each extreme flood scenario is like in Shanghai (e.g., their discharge or precipitation)? | We have rewritten and enhanced the description of the different flood scenarios in Shanghai in section 1 "Introduction" (p.4. Line 47-61) and discharge/inundation of different extreme flood scenarios from the previous study in in section 5.2 "Future challenge and adaptation strategies" (p.31. Line 308-315). |
| | What is the implication of this study to other cities or future research? | Other cities should also pay attention to risk analysis and management of low probability-high impact flood events. The method of this study could be useful for other cities. In addition, the estimation of different building damages could inform future flood damage studies to consider various assets with more precise evaluations. |
| | Have you considered validating your simulated results or comparing them with other Shanghai flood risk assessments? | Thank you for pointing this out. Other Shanghai flood risk assessments have different methodologies and assessment objectives. We have validated our results with Wu et al. (2019) and Shan et al. (2019) who have building and residential building flood risk assessments for Shanghai (p.28. Line 278-281) respectively. |
| Specific comments | Line 39. Why do you think Shanghai 'should increasingly install flood protection, with a focus on hard measures'? Please can you justify or provide evidence? | Thank you for this point. This sentence caused a misunderstanding, and we deleted it. We revised it in the section 1 "Introduction", the seawalls and levees can be destroyed because of the multiunit structures' standards used during the long temporal construction process and the historical crest height in the Huangpu River growing from 1950 to 2000. This is also the reason we focus on low probability-high impact flood scenarios (p.3. Line 37-46). |
| | Line 53. Please explain what a two-dimensional MIKE 21 flow model is and its features as part of the introduction. | Thank you for pointing this out. As you suggested, we provide a better explanation on what a two-dimensional MIKE is in the new section 3.1 "Flood hazard modeling" (p.9. Line 106-111). |
| | Line 68. I agree with the authors that "Accurate loss data play an integral role in assessing the damages of buildings. But obtaining accurate data is a challenge shared in many areas (Middelmann-Fernandes, 2010), especially in assessing the damage of buildings." However, the challenge of obtaining accurate loss data is not the focus of this manuscript or hasn't been solved by this study. Therefore, I don't think they are directly relevant as part of the introduction. Consider | As suggested by the reviewer, "Accurate loss data play an integral role in assessing the damages of buildings. But obtaining accurate data is a challenge shared in many areas (Middelmann-Fernandes, 2010), especially in assessing the damage of buildings.", is not the coherent of this manuscript and so we have deleted this clause. |

| | | |
|---|---|---|
| moving it to the methodology section. | | |
| Line 95. What does 'construction industry value' mean? | Thank you for pointing this out. The conception of 'construction industry value' from the Shanghai statistical yearbook and means the total value of the construction industry. The construction industry contains various buildings in Shanghai, including residential buildings, office buildings, commercial buildings, and others. | |
| Line 102. Three types of models were developed for the assessment, including atmospheric models, ocean models, and coastal models. Consider placing them in the methodology section instead of the data section. Again, more information about these models is expected. | Thank you for the suggestion. We revised accordingly (p.8. Line. 107-125). | |
| Line 126, Table 1. How were the Average Construction Costs calculated? Were different building types weighted? Why is the Average Construction Cost (1157) smaller than the lower bound of the range (1228) for commercial buildings? | Thank you for bringing this to our attention. The average construction cost calculates by the mean value of each construction cost of building type. The comment is correct, the average commercial construction cost was incorrect. Since the inaccurate number of the average construction cost was used, the asset value of buildings, damage estimation, and risk evaluation have all been recalculated. The Average Construction Cost for commercial buildings is updated to 1407 USD/m². The revised text reads as follows on: | |

Table 1. Common construction costs of various buildings in Shanghai.

| Building Type | | Construction Cost (USD/m² CFA) | Average Construction Cost (USD/m²) |
|---|---|---|---|
| Residential | Apartments, high rise, average standard | 668-740 | 874 |
| | Apartments, high rise, high end | 1554-1697 | |
| | Terraced houses, average standard | 446-477 | |
| | Detached houses, high end | 666-740 | |
| Commercial | Retail malls, high end | 1228-1585 | 1407 |
| Office | Medium/high rise offices, average standard | 868-1156 | 1157 |
| | High rise offices, prestige quality | 1158-1445 | |
| Industrial | Industrial units, shell only (Conventional single story framed units) | 432-540 | 486 |

We revised the table in section 3.2.1 "Asset value of building" (p.12.Line 144).

| | |
|---|---|
| Line 154. The 'W' in 'Where' should be in lowercase. | Thank you for pointing this out. However, the 'W' is not the 'W' in 'Where'. The 'W' is a representative variable. |
| Line 154. What does 'surface area of building' mean? Does it include the wall as well? Line 164, Equation 2. f(x) means a function at an element x. However, an x is missing on the right to the equal sign. Please | Thank you so much for taking the time to write such a thorough review. We have rewritten the functions and have given examples on 'building asset value' (p.13. Line 160) and 'the damage values' (p.15. Line 177). |

| | modify the equation and explain what x means. Line 175, Equation 3. More information is needed to explain Equation 3. | |
|---|---|---|
| | Line 178. Explain Getis-Ord. | Thank you for addressing this point. We combine the Getis-Ord Gi* to the new section "materials and method". The revised text reads on (p.17. Line 187-189). |
| | Line 199. Is the building asset value for the first floor of all four building types? | The building asset value is calculated not only for the first floor of all four building types, but for all floors of all building type (p.13. Line 160). |
| | Line 207, Figure 4 (also Line 231, Figure 5 and Line 251 Figure 6). Since the Average Construction Cost is used for each of the four building types, is it true in Figures 4-6 that the buildings with higher 'Building Asset Values' are buildings taking a larger land area? | The answer is yes if we compare 'Building Asset Values' in the same type of building. The reason for this is that the construction costs for the same type of building are the same and the variable is only the surface area of one building. On the other hand, when we compare the 'Building Asset Values' in different types of buildings, the answer is no. Because there are two variables, one is the cost of construction, and the other is the building's surface area. |
| | Line 329, Table 5. Table 5 provides a comparison of flood adaptation measures in Shanghai. However, how can these measures, especially the soft ones, be reflected in the simulations? The simulation results and the soft adaptation measures are disconnected, and more discussion is needed here. | Thank you for pointing this out. Table 5 provides a comparison of flood adaptation measures in Shanghai, but it is not calculated in the simulations. |
| colspan Review 2 | | |
| General comments | Building damage assessment is very important in urban flood risk management. This study presents an assessment of possible exposure and damage losses of buildings in Shanghai. The topic of this study is valuable, and However, the quality and innovation of the current manuscript are not satisfactory. First of all, lots of figures are poor in quality and hard to read. Besides, the building damage assessment method used in this study lack of innovation. In any case, I have a few recommendations that I believe will help the authors to clarify their contribution and improve the readability of the text in a few passages. | We are grateful for the reviewer's general comment on the study. Flood risk assessment approaches are widely implemented in many regions and the approaches themself have been well developed. This study improved the flood risk assessment in three aspects: (1) The integrated flood model that simulated extreme compound flood scenarios was first produce in Shanghai. (2) The building flood risk in this study is a clear enough future for Shanghai. (3) The building flood damages, risk, and risk's spatial patterns were for the first time evaluated in the whole city (except the islands) on fine scale (community and district level).

We further re-edited the figures to make them easier to read. |
| Specific comments | More information on the urban flood modelling by extreme flood scenarios caused by storm surges, precipitation, and fluvial floods, should be provided | Thank you for this point. The methodology and details of the flood scenario have been published by the co-authors in the paper Wang et al 2019. We revised our manuscript and clarified the scenarios in the methodology. The revised text reads as follows (p.9. Line 107-125). |

| | | |
|---|---|---|
| | in the study. For example, what is the detailed combination of storm surges, precipitation, and fluvial floods. | |
| | Most of the figures in the manuscript are very poor in quality and hard to meet the standard for this journal, such as Figs. 5, not clear enough. | Thank you for this point, we re-export the figure and improved the quality with clearer colors. |
| | Table 5 presents comparison of flood adaptation measures in Shanghai, how does it make any sense? Anyway, the discussion in this study seems meaningless. | Sorry that the discussion section did not well present its values. The discussions attend to address the threat of extreme flood events and their simulation results. We also narrate the potential flood adaption techniques and the discrepancy between the master plan and the academic result. The discussion would be helpful in providing information to the decision-makers and a statement for the researcher to simulate the flood scenario in Shanghai in the future. We significantly revised the section and concentrate on discussing two aspects: (1) Section 5.1 analyzed the uncertainty and limitations of the study, and further analyzed the direction to enhance the model performances. The suitability of transferring the model to other study areas is also discussed (p.26. Line 265-297). (2) Section 5.2 discusses future challenge and adaptation strategies in Shanghai (p.29. Line 298-331). |
| | The building flood damage assessment method used in this study is too simple and lacks the novelty. | Thank you for bringing this up. We revised the revised text reads as follows (p.9. Line 107-125). |
| | Should be Figure 7 and Figure 8 instead of Fig. 7, Fig. 8 in Page 14. | We have checked all figures and tables in the manuscript and updated them on this point. |
| | The methods of assessment of building damages in extreme floods used in this study are mainly derived from existing studies, thus what is the main contribution of this study. | Thank you for the comment. We revised "Materials and methods" (p.8. Line. 98-194). |

---

## Author Response (AR2)

|  | Comments | Submitted Answer |
|---|---|---|
| | Editor | |
| General comments | Many thanks for revising the paper. You have convincingly addressed the critical points raised in the first review round. The paper brings unique empirical evidence and combines state-of-the art flood assessment with damage modelling. I now only have two minor issues as raised by reviewer 2:

Pls. specifiy more concretely the differences of this paper from earlier contributions and stress more clearly its additional value and contribution. | We are very grateful for the editor's kind and helpful comments.
There has been already research in this field with several papers published in the past focusing on Shanghai, specifically Wu et al. (2019) and Shan et al. (2019). However, it's important to note that the specific objectives under flood risk differed between these studies: In the study by Wu et al. (2019), the hazard was defined as a 1/10,000-year fluvial flood scenario (river flood from the Huangpu River), whereas we use several hazard scenarios to cover a broader range of low probability – high impact flood events. For exposure, the building asset value in Shanghai was calculated based on the building floor area, which was obtained from the building floor area per district, and the population density per sub-district. Our approach is based on this method however with improvements in spatial resolution and methodology. We adjusted section 5.1 to make this clearer. In another study by Shan et al. (2019), which focused on the same hazard and vulnerability subjects, exposure was determined by considering residential buildings and household properties, using the market price of residential buildings. We explain the advantages of our approach compared to this method in "section 5.1 Uncertainties and limitations" (Line 286-295[1]). |
| Other general comment | In respect to the applicability of the methodological approach to other mega cities, I wonder whether you could discuss context-specific challenges of doing this. | Thank you for this point.
Many studies address the three subjects, hazards, exposure, and vulnerability in risk assessment, however, at the same time, due to the different subjects assessed, the resulting risk estimates differ. It is important to clearly specify the subject or subjects under discussion in the flood risk assessment, as one or many subjects change can alter the objectives. This answer is also an echo to the last question, that the different between the flood risk assessment for the same city (Line 295-301).
Another major challenge is the availability of data: we have access to a rich dataset and extensive research on hazard, exposure, and vulnerability, allowing us to undertake risk analysis chain in Shanghai. This availability of resources can pose a challenge for other cities that may have limited data and research in conducting their own risk assessments. We revised this point in "Section 5.1 Uncertainties and limitations" (Line 317-319). |
* * *
[1] The line numbers in the author's response match the manuscript track-changes file.

| Referee #3 | | |
|---|---|---|
| General comments | The paper presents a flood risk assessment for Shanghai, providing an indication of the potential damage of different building types in low-probability/high-impact flood scenarios in a well-written manner.

While, given the speed with which climate change is progressing, analysing extreme case scenarios can be considered a valuable addition to the field of flood risk assessment, it remains unclear which additional contributions the paper makes. This becomes particularly apparent, as very similar papers have been published in the past, using the same methods and also focussing on Shanghai (Wu et al., 2019, Shan et al., 2019). I would recommend that the authors further differentiate this paper from these two previous ones by clearly articulating its additional value and contribution. | Thank you so much, Franziska, for your kind comments.
As the question also concluded with Editor:
There has been already research in this field with several papers published in the past focusing on Shanghai, specifically Wu et al. (2019) and Shan et al. (2019). However, it's important to note that the specific objectives under flood risk differed between these studies: In the study by Wu et al. (2019), the hazard was defined as a 1/10,000-year fluvial flood scenario (river flood from the Huangpu River), whereas we use several hazard scenarios to cover a broader range of low probability – high impact flood events. For exposure, the building asset value in Shanghai was calculated based on the building floor area, which was obtained from the building floor area per district, and the population density per sub-district. Our approach is based on this method however with improvements in spatial resolution and methodology. We adjusted section 5.1 to make this clearer. In another study by Shan et al. (2019), which focused on the same hazard and vulnerability subjects, exposure was determined by considering residential buildings and household properties, using the market price of residential buildings. We explain the advantages of our approach compared to this method in "section 5.1 Uncertainties and limitations" (Line 286-295). |
| | Generally speaking, the paper is empirical in nature and, so far, is missing a well-developed theoretical and conceptual foundation (and chapter). Positioned between physical (flood assessment) and economic geography (asset damage assessment / flood management), it has not yet been given a real focus for its conceptual contribution. While interdisciplinary research has its merits, it would be helpful if the authors could clearly articulate which part of the work is linked to which area of the field and engage more strongly with relevant literature. | Thank you for this point.
The strength and innovative part of our paper is the (equal) combination of exposure assessment and vulnerability analysis. In order to make our interdisciplinary approach and conceptual framework clearer, we have revised "Section 1 Introduction" in order to align our concept and adapt the risk concept framework by the IPCC (Line 23-25). Based on the conceptual framework and adapt to the low probability-high impact flood scenarios in Shanghai, we have developed this study (See Line 80-82). |
| | As an economic geographer, I am unable to comment in greater detail on the flood assessment part of the paper. However, given the papers aim of providing a foundation for scenario-based decision-making, cost-benefit analysis, and flood risk management, I believe that it could be more strongly integrated into discussion in the field of urban planning. For instance, the contribution of the paper could be strengthened by incorporating Shanghai's current flood management measures into the GIS analysis.
• At which point will the current measures fail to protect?
• How would these measures have to be adapted to reduce potential | Thank you for your kind comments.
We improved several sections to address your comments:
• The present protection level of the levees along the Huangpu River for the lowest sections is around 1/50-year flooding scenario (Ke et al., 2018) (Line 42-46).
• Thank you for this point. Table 6 provides potential measures could be adapted in Shanghai, but it is not calculated in the flood risk assessment.
• For instance, in its Master Plan 2017-2035, Shanghai is going to further develop its five new district centers at Jiading, Songjiang, Qinpu (Songjiang-Qingpu low-lying area), Fengxian and Nanhui. These five |

| | damage? How much would this cost?
• Are there areas which should be prioritised when implementing new protective measures?
• Are there areas that could not be protected in an economically (cost-benefit) sustainable way? Implementing such discussions would truly increase the real-world impact of the paper. | district centers are planned to be nodal areas in Shanghai and provide more public services for the growing population. However, based on our findings, the Songjiang-Qingpu low-lying area protected by levee with 1/50-year flooding scenario along the Huangpu River, is a hot spot of flood damages. In recognition of this vulnerability, local stakeholders have acknowledged the necessity of implementing levee enhancements in the Songjiang-Qingpu low-lying area. Therefore, future flood protections in these locations, particularly the drainage system and the building structures, must be designed to a higher standard (Line 350-353).
• Thank you for this point. As of now, we have not conducted a cost-benefit calculation for buildings in Shanghai. However, we acknowledge the potential for such analysis to be explored in future research. |
|---|---|---|
| | Finally, the authors claim that their study can be replicated in other megacities. However, an in-depth discussion of the context-specific challenges of conducting the study and how they might differ and, consequently, have to be addresses differently in other megacities is missing so far. Providing a list of recommendation for other researcher and urban planner who would like to replicate this study in other megacities would be a potential additional contribution to research and planning practices. | Thank you for this point.
This work can be replicated in other megacities by adhering to two guidelines. Firstly, it is crucial to clearly define the risk assessment objectives. Secondly, adequate datasets, such as flood hazard maps, or data for calculating asset value for buildings, infrastructure, etc. should be prepared to facilitate the risk assessment process. The awareness and challenge with adhering to these guidelines are discussed in the other general comment from Editor. |

---

## Author Response (AR3)

Dear Editor,

We appreciate the chance to let us improve the English in the paper, our native speaker co-author carefully improved the language. Please feel free to find the updated manuscript.

With very best wishes,

Jiachang (on behalf of co-authors)